# The Plastic Deformation Mechanisms of hcp Single Crystals with Different Orientations: Molecular Dynamics Simulations

**DOI:** 10.3390/ma14040733

**Published:** 2021-02-04

**Authors:** Zhi-Chao Ma, Xiao-Zhi Tang, Yong Mao, Ya-Fang Guo

**Affiliations:** 1Institute of Engineering Mechanics, Beijing Jiaotong University, Beijing 100044, China; 15115314@bjtu.edu.cn (Z.-C.M.); xztang@bjtu.edu.cn (X.-Z.T.); 2School of Materials and Energy, Yunnan University, Kunming 650091, China; maoyong@ynu.edu.cn

**Keywords:** hcp metals, plastic deformation, orientation effect, molecular dynamics simulations

## Abstract

The deformation mechanisms of Mg, Zr, and Ti single crystals with different orientations are systematically studied by using molecular dynamics simulations. The affecting factors for the plasticity of hexagonal close-packed (hcp) metals are investigated. The results show that the basal <*a*> dislocation, prismatic <*a*> dislocation, and pyramidal <*c* + *a*> dislocation are activated in Mg, Zr, and Ti single crystals. The prior slip system is determined by the combined effect of the Schmid factor and the critical resolved shear stresses (CRSS). Twinning plays a crucial role during plastic deformation since basal and prismatic slips are limited. The $\left\{10\bar{1}2\right\}$ twinning is popularly observed in Mg, Zr, and Ti due to its low CRSS. The $\left\{10\bar{1}1\right\}$ twin appears in Mg and Ti, but not in Zr because of the high CRSS. The stress-induced hcp-fcc phase transformation occurs in Ti, which is achieved by successive glide of Shockley partial dislocations on basal planes. More types of plastic deformation mechanisms (including the cross-slip, double twins, and hcp-fcc phase transformation) are activated in Ti than in Mg and Zr. Multiple deformation mechanisms coordinate with each other, resulting in the higher strength and good ductility of Ti. The simulation results agree well with the related experimental observation.

## 1. Introduction

Because of the low symmetry and a limited number of activated slip systems, hexagonal close-packed (hcp) metals present a deformation behavior which is quite different from that of materials with cubic crystalline structure. In hcp metals, the most popular plastic deformation mechanisms are slip and twinning. In general, slip is always along the lattice close-packed direction on the close-packed plane. Thus in hcp metals, slip system varies with the change of the c/a ratio. For example, basal <*a*> slip is most easily activated in hcp Mg [1]. However, prismatic <*a*> slip dominates the plasticity of hcp Ti and Zr [2,3]. Under the c-axis loading when the basal and prismatic slips are restricted, the activation of pyramidal <*c* + *a*> slips is possible [4,5,6].

Twinning is popularly observed in hcp metals. Compared to metals with cubic crystal structure which have limited numbers of twinning systems, there are at least seven twinning modes in hcp metals, such as $\left\{10\bar{1}1\right\}$, $\left\{10\bar{1}2\right\}$, $\left\{10\bar{1}3\right\}$, $\left\{11\bar{2}1\right\}$, $\left\{11\bar{2}2\right\}$,$\;\left\{11\bar{2}3\right\}$, and $\left\{11\bar{2}4\right\}$, twins. The lattice reorientation induced by twinning can provide the possibility of the activation of more slip systems, which may be favorable for further deformation. Factors influencing the twinning modes include the *c*/*a* ratios, lattice packing densities, interplanar spacings, stacking fault energies, and so on. In Mg and its alloys, $\left\{10\bar{1}2\right\}$ tension twin and $\left\{10\bar{1}1\right\}$ compression twin are the most popular twinning modes [7,8,9,10]. In Ti and its alloys, the most commonly observed tension twins are $\left\{10\bar{1}2\right\}$ and $\left\{11\bar{2}1\right\}$, and the compression twins are $\left\{11\bar{2}2\right\}$ and $\left\{10\bar{1}1\right\}$ [11,12,13,14]. Twinning-induced plasticity (TWIP) has been put forward to interpret the abnormal high ductility of Ti at low temperature because of the rich twinning system in Ti [15]. In Zr and its alloys, $\left\{10\bar{1}2\right\}$ tension twin is more popularly observed than $\left\{10\bar{1}2\right\}$ twin [16], whereas $\left\{11\bar{2}2\right\}$ twin is dominant during c-axis compression. At high temperatures, $\left\{10\bar{1}1\right\}$, twin is also activated under compression in Zr [17,18]. Moreover, different types of double twins (DT) are observed in the deformation of hcp metals; for example, $\left\{10\bar{1}1\right\}-\left\{10\bar{1}2\right\}$. DT is particularly associated with the plastic deformation and premature failure of Mg alloys [19], while $\left\{10\bar{1}1\right\}-\left\{10\bar{1}1\right\}$ double contraction twins (DCTWs) have also been explored in Mg alloys, recently [20]. $\left\{11\bar{2}1\right\}-\left\{11\bar{2}2\right\}$ DT was observed at lower temperatures in hcp Ti [21].

In addition to slip and twinning, phase transformation plays an important role on the plasticity of hcp metals. It is generally known that allotropic transformation occurs at 882 °C in pure Ti from hcp ɑ-Ti to bcc β-Ti [22]. In addition, the fcc-Ti was observed in thin films of Ti [23] and in Ni/Ti [24] and Al/Ti multilayers [25]. The related studies suggested that the hcp-fcc phase transformation in Ti is induced by applied stress rather than thermal activation [26]. The surface and orientation effects on stress-induced hcp-fcc phase transformation in Ti nanopillars has also been investigated, recently [27]. The biphasic HCP/FCC phase boundary are theoretically researched using classic molecular dynamic simulation [28].

In general, the hcp metals display strong anisotropic mechanical behavior because of their asymmetric crystallographic structure. The orientation effect has a great effect on deformation behavior of such asymmetric structure [29,30,31,32]. Different deformation mechanisms can be activated under different loading conditions. In Mg and its alloys, the <*c* + *a*> slips are activated when the basal slip is restricted, and different tension and compression twins appear under the c-axis tension and compression, separately. By using molecular dynamic simulations, Zu et al. [29] have reported the orientation effect on the initial plastic deformation of Mg single crystals, the competition mechanism of different plastic deformation under different loading conditions was discussed. In polycrystalline Zr, the orientation effect on micromechanical responses have been studied using spherical nanoindentation by Pathak et al. [30]. Battaini et al. [31] have reported the orientation effect on the mechanical behaviors of pure Ti plate, and the flow stress varied greatly due to the sample’s orientations. 

Because of the complexity of deformation mechanisms in hcp metals, clarifying the affecting factors for the strength and plasticity is of great importance for improving the mechanical properties of hcp materials. In this paper, molecular dynamics simulations are employed to investigate the deformation mechanisms of different hcp single crystals, such as Mg, Zr, and Ti, with different *c*/*a* ratios under different loading conditions. By comparing the deformation mechanisms of different hcp metals and analyzing the orientation effect on deformation behaviors, the affecting factors for the plasticity of hcp metals are further discussed.

## 2. Simulation Method

In order to study the plastic deformation mechanisms under different loading conditions, two single crystal models (A and B) with different orientations are established. Figure 1 shows the perfect single crystal models with square cross sections. The side length of the samples is about 14.3 nm and the height-to-side ratio is about 2:1. The number of atoms in the systems is about 280,000. The free boundary conditions are applied in the x- and z-directions, while the fixed displacement boundary condition is assigned to the y-direction. The loading direction is always along the y-direction. In Figure 1a, we orientate the $\left[1\bar{2}10\right]$ direction as x-axis, the $\left[0001\right]$ direction as y-axis, and $\left[\bar{1}010\right]$ direction as z-axis for the starting structure. Then the crystal is rotated around the $\left[\bar{1}010\right]$ axis from 0° to 90°, which means that the angle (θ) between the loading direction (y) and the c-axis ranged from 0° to 90°. In Figure 1b, x-, y-, and z-axis respectively parallel the $\left[\bar{1}010\right]$, $\left[0001\right],$ and $\left[1\bar{2}10\right]$ directions in the starting structure. Then the crystal is rotated around the $\left[1\bar{2}10\right]$ axis from 0° to 90°. The angle (β) between the loading direction (y) and the c-axis ranged from 0° to 90°. Ten different angles θ of 0°, 10°, 15°, 32.1°, 43.3°, 51.6°, 70°, 75°, 80°, 90° and twelve different angles β of 0°, 10°, 15°, 28.6°, 35.9°, 47.4°, 55.4°, 58.6°, 70°, 75°, 80°, 90° are examined in our simulation. The simulations are performed in constant NVT ensemble with a velocity Verlet integrator. The temperature is controlled at 5 K. The EAM potentials of Sun et al. for Mg [33], Mendelev and Ackland (MA) for Zr [34], Ackland et al. for Ti [35] are used. The default calculation method in LAMMPS program [36] which is based on the classical statistical mechanics [37,38] and the virial theorem [39] is applied to calculate the stress. The AtomEye software [40] is used for visualizing the evolution of the atomistic structures. The crystal defects are colored by the Ackland [41] and Common Neighbor Analysis (CNA) [42].

At first, the perfect crystal is relaxed for 30 ps (5000 time steps) at zero force to minimize the potential energy. Then a uniaxial tension is applied along the *y*-direction with a constant strain rate of about $5\times 10^{7}\;s^{-1}$ on the 1.5-nm-thick top layer, while the 1.5-nm-thick bottom layer is fixed. The system is relaxed for 6 ps before the next increment of tensional displacement is applied. The maximal strain obtained in our simulation is about 10%.

## 3. Results and Discussion

### 3.1. The Initial Plastic Deformation Mechanisms of Mg, Zr, and Ti

The plastic deformation of Mg, Zr, and Ti single crystals under uniaxial tension with different orientations are shown in Table 1 and Table 2. The initial plastic deformation mechanisms in the strain range of 0.2% over the yield point are indicated. The dislocation, crystallographic reorientation, and hcp-fcc phase transformation are identified as or, blue, and green, respectively. The dislocation mechanisms include the basal <*a*> dislocation (BD <*a*>), prismatic <*a*> dislocation (PrD <*a*>), pyramidal <*c* + *a*> dislocation (PyD <*c* + *a*>), shear band (SB), basal stacking fault (BSF), and the cross-slip of prismatic *<a>*, and pyramidal <*a*> dislocation. Crystallographic reorientation includes twinning and basal/prismatic (BP) transformation. Several twinning modes are observed, including $\left\{10\bar{1}1\right\}$ and $\left\{10\bar{1}2\right\}$ twins, $\left\{10\bar{1}1\right\}-\left\{10\bar{1}2\right\}$ and $\left\{11\bar{2}1\right\}-\left\{11\bar{2}2\right\}$ double twins. It can be seen in Table 1 and Table 2 that the types of plastic deformation mechanisms in Ti are more than in Mg and Zr. The cross-slip of prismatic *<a*> and pyramidal <*a*> dislocation (dark brown), $\left\{11\bar{2}1\right\}-\left\{11\bar{2}2\right\}$ double twins (black blue), the fcc-Ti phase transformation (green) appear in Ti. We will present the details of the different plastic deformation mechanisms in Mg, Zr, and Ti single crystals in the following sections.

### 3.2. Slips

In hcp metals, the dominant slip direction is always the lattice close-packed direction $\left[1\bar{2}10\right]$. The favored slip plane in Mg is the basal plane, while favored slip plane in Ti and Zr is the prism plane. The selection of the activated slip modes in hcp metals subjected to an applied stress is dependent on the Schmid factor and the critical resolved shear stress (CRSS) according to the Schmid law. Table 3 and Table 4 present the Schmid factors for the four slip systems in Mg, Zr, and Ti when the tension is along the direction with different θ or β. The slip system with a larger Schmid factor is predicted to be preferentially activated during deformation. As shown in Table 3 and Table 4, when the tension is applied along the *c*-axis, the pyramidal <*c + a*> dislocations are favored because the Schmid factors of basal and prismatic slips are zero. With the increase of θ or β, the Schmid factor of basal slip becomes larger, thus basal slips gradually dominate the initial plastic deformation. When θ or β approaches to 90°, the prismatic <*a*> slip is favored than the basal <*a*> slip according to the Schmid factor, thus the prismatic <*a*> dislocations are the main plastic deformation mechanism when θ ≥ 70°.

Figure 2 shows the plastic deformation of Zr single crystal with different orientations (θ). Figure 2a shows the results at θ = 15°. The atoms on perfect hcp lattice are not shown. When the applied strain is up to 5.75%, a $\left\{10\bar{1}1\right\}$ pyramidal dislocation (f = 0.493) nucleates predominately (Figure 2(a1)). With increase of the strain, a SB appears caused by the $\left\{1\bar{2}11\right\}$ pyramidal slip (f = 0.453) (Figure 2(a2)). The lattice reorientation with the rotation angle of 34° is induced in this pyramidal SB, while the rotation angle is just the same as the misorientation angle of $\left\{1\bar{2}11\right\}$ twin. This deformation mode was usually defined as type II $\left\{1\bar{2}11\right\}$ twin in previous works [43,44]. The detailed analysis of SB has been applied in Mg by Zu et al. [29], which is just the same as the results in this simulation.

When θ is up to 32.1°, the basal <*a*> dislocation dominates the initial plasticity of Zr. As shown in Figure 2(b1), a leading dislocation in basal plane nucleates at the free surface, followed by a trailing dislocation. Meanwhile, a BSF exists between the leading and trailing partial dislocations. In Figure 2(b2), the trailing dislocation catches up with the leading to fulfill the basal <*a*> dislocation. Then the basal <*a*> dislocation moves on and extends to the free surface, with a slip step left on the surface (Figure 2(b3)). Snapshots in Figure 2(c1,c2) are the deformation of Zr at θ = 90° when the strains are 5.62% and 5.86%, respectively. When the tension strain is 5.62%, the prismatic <*a*> slip (f = 0.433) is primarily activated (Figure 2(c1)). With the increase of strain, new prismatic <*a*> dislocations successively nucleate and extend across the sample (Figure 2(c2)).

In Figure 2b, the basal <*a*> dislocation dissociates into the Shockley partials on the basal plane. However, the perfect prismatic <*a*> dislocation can exist without any extensive stacking fault (SF) generated in Figure 2c. The key factors determining the slip mode of dislocation are Peierls stress and stacking faults energy (SFE). In Zr and Ti, the slip of a prismatic dislocation has minimal Peierls stress [45,46], indicating that the full prismatic <*a*> dislocation is easier to activate than dislocations in other plane. The magnitude of SFE determines the ease of dissociation into partials. The basal dislocation is easier to split into partials because of its lower SFE [47]. Therefore, the dissociation of basal <*a*> dislocation into two partials and a BSF are usually observed.

The simulation results of the Mg single crystals are primarily consistent with those of the Zr for model A, whereas the simulation result of Ti single crystal is somewhat different when θ ≥ 70°. The prismatic <*a*> dislocations in Ti can cross-slip into an intersecting pyramidal plane. The activation of secondary slip system is helpful for achieving a complete three-dimensional deformation, which also promotes nucleation of subsequent $\left\{10\bar{1}1\right\}$ twins (the detail will be discussed in Section 3.3.2). The similar cross-slip of <*a*> dislocations in Ti was also observed by Caillard et al. [48].

We notice that in Table 4 when β = 10°, the Schmid factor of the $\left\{10\bar{1}1\right\}\langle 11\bar{2}3\rangle$ pyramidal slip (f~0.470) is much higher than that of the basal <*a*> slip (f = 0.125) in Mg and Ti. However, the basal <*a*> slip occurs prior to the pyramidal <*c + a*> slip because of the higher CRSS for the <*c + a*> slip at low temperatures. The CRSS is a material parameter which is dependent on deformation temperature, strain rate, impurity, etc. Chapuis et al. found that at the temperature of 298 K for Mg, the CRSS values are 5 MPa for the basal <*a*> slip and 110 MPa for the first order pyramidal <*c + a*> slip [49]. It indicated that the CRSS value for pyramidal <*c + a*> slip is about 22 times larger than that for basal <*a*> slip. Thus the basal slip is easier to be activated than the pyramidal slip.

In our simulations, except for the SBs caused by the $\left\{11\bar{2}1\right\}$ pyramidal slip, only $\left\{10\bar{1}1\right\}$ pyramidal slip is observed in Mg. This is different from the experimental observation that $\left\{11\bar{2}2\right\}$ pyramidal slip dominates the plasticity in Mg. Tang et al. [50] have explained this discrepancy between experiment and simulation. In molecular dynamics simulations, they have observed the $\left\{10\bar{1}1\right\}$ pyramidal <*c* + *a*> dislocation first during *c*-axis loading by sequential nucleation of leading and trailing partial dislocations. Subsequently, the formation of $\left\{11\bar{2}2\right\}$ pyramidal dislocations is achieved either by cross-slip of screw-segments or by cooperative slip of edge-segments on two equal and alternative $\left\{10\bar{1}1\right\}$ pyramidal planes. Afterwards, slip occurs predominantly on $\left\{11\bar{2}2\right\}$ pyramidal planes, which is consistent with experiments. Wu et al. also explained this phenomenon through the viewpoint of the energetics of dislocation transformations [48]. The energy difference between <*c* + *a*> dislocations on $\left\{10\bar{1}1\right\}$ and $\left\{11\bar{2}2\right\}$ planes in Mg is small, suggesting relatively easy cross-slip. But for Zr and Ti, the <*c* + *a*> dislocation is energetically preferable on $\left\{10\bar{1}1\right\}$ pyramidal plane. Therefore, the <*c* + *a*> slip should dominate on the $\left\{10\bar{1}1\right\}$ pyramidal plane and cross-slip is difficult because of the large energy differences. However, some experiments show the cross-slip of $\left\{11\bar{2}2\right\}$ pyramidal <*c* + *a*> dislocations in Ti and its alloys [51,52], especially under *c*-axis compression at high temperatures. In these experiments, the activation of $\left\{11\bar{2}2\right\}$ pyramidal slip and cross-slip may be related with the stress, temperature, solid solution alloying, or a combination of them.

### 3.3. Twinning

Twinning is a major deformation mode in hcp metals. Four twinning modes are most commonly observed, including two extension twins $\left\{10\bar{1}2\right\}\langle \bar{1}011\rangle$ and $\left\{11\bar{2}1\right\}\langle \bar{1}\bar{1}26\rangle$, two compression twins $\left\{11\bar{2}2\right\}\langle 11\bar{2}3\rangle$ and $\left\{10\bar{1}1\right\}\langle 11\bar{2}3\rangle$. In our simulation, $\left\{10\bar{1}2\right\}$ twins appear in Mg, Zr, and Ti under *c*-axis tension. $\left\{10\bar{1}1\right\}$ twins are observed in Mg and Ti. $\left\{1\bar{2}11\right\}$ type II twins (SBs) appear in Mg, Zr, and Ti. Moreover, $\left\{10\bar{1}1\right\}-\left\{10\bar{1}2\right\}$ double twins of Mg and $\left\{11\bar{2}1\right\}-\left\{11\bar{2}2\right\}$ double twins of Ti are found in several loading directions.

#### 3.3.1. $\left\{10\bar{1}2\right\}$ Twinning

From Table 1 and Table 2, it is shown that the $\left\{10\bar{1}2\right\}$ twins are observed when θ = 0° (*c*-axis tension) for Mg, Zr, and Ti. The CRSS of $\left\{10\bar{1}2\right\}$ twins in Mg, Zr, and Ti are 30 MPa, 165 MPa, and 260 MPa, respectively [49,53,54], which are less than the values of other twins. Therefore, the $\left\{10\bar{1}2\right\}$ twins are more active in hcp metals, especially in Mg. $\left\{10\bar{1}2\right\}$ twinning has been popularly observed by experiments and atomistic simulations, and the mechanism of $\left\{10\bar{1}2\right\}$ twinning has been widely discussed [7,8,10,11,16]. Figure 3 shows the initial plasticity of Zr single crystal under *c*-axis tension. The pyramidal <*c* + *a*> dislocation occurs first, then the basal slip follows (Figure 3a,b). In the subsequent deformation, the reoriented crystal due to the BP transformation and $\left\{10\bar{1}2\right\}$ twinning runs cross the sample and extends toward the terminal of crystal (Figure 3c,d). The BP transformation and $\left\{10\bar{1}2\right\}$ twinning dominate the plasticity of Zr when θ = 0°. Figure 3e,f presents the magnified view of the terraced interface, which consists of BP interfaces, $\left\{10\bar{1}2\right\}$ coherent twinning boundaries (CTBs), and interface defects. Two-layer twinning dislocations ($b_{2}^{\mathrm{CTB}}$), two-layer BP disconnections ($b_{2}^{\mathrm{BP}}$), and one-layer BP disconnections ($b_{1}^{\mathrm{BP}}$) are found at the interface. The growth of the reoriented crystal is achieved by the migration of CTBs and BP interfaces via the movement of interface defects, accompanying with the transformation between $\left\{10\bar{1}2\right\}$ CTB and BP interface. The deformation mechanism of Zr single crystal under *c*-axis tension in this simulation is almost the same as that of Mg single crystal obtained by Zu et al. [29].

For Ti single crystal under *c*-axis tension, the situation is somewhat different from that of Mg and Zr. As shown in Figure 4a, the $\left\{10\bar{1}2\right\}$ twin nucleates directly at the free surface, with no pioneering dislocations observed. When the strain increases, the twin region continues to expand (Figure 4b). Then the BSFs are emitted at the BP interface (Figure 4c), which caused the formation of fcc-Ti in Figure 4d. The formation of fcc-Ti is induced by the accumulation of BSFs, which is consistent with the analysis of Li et al. [55]. Figure 4e–h shows the nucleation and growth of $\left\{10\bar{1}2\right\}$ twin at different time steps viewed along the $\left[1\bar{2}10\right]$ direction. The $\left\{10\bar{1}2\right\}$ twin nucleates at the free surface in Ti single crystal (Figure 4e). The BP interfaces emerge by the transformation of the CTBs (Figure 4f). Then the BP interfaces migrate via one-layer and two-layer disconnections (Figure 4g,h).

#### 3.3.2. $\left\{10\bar{1}1\right\}$ Twinning

The $\left\{10\bar{1}1\right\}$ compression twins are observed when θ or β ≥ 70° for Mg and θ ≥ 70° for Ti. The CRSSs of $\left\{10\bar{1}1\right\}$ twin in Mg, Zr, and Ti are 55 MPa, 300 MPa, and 360 MPa, respectively [49,53,54]. Therefore, the $\left\{10\bar{1}1\right\}$. twins are most easily formed in Mg. No $\left\{10\bar{1}1\right\}$ twin is observed in Zr because of the higher CRSS. Many researchers reported that the $\left\{10\bar{1}1\right\}$ twins can only form at elevated temperatures in Zr [17,18]. For Ti, the nucleation of $\left\{10\bar{1}1\right\}$ twins is usually related to the cross-slip of the dislocations [13].

In order to understand the nucleation and growth mechanisms of $\left\{10\bar{1}1\right\}$ twin, the microstructural evolutions of Mg and Ti sample when θ or β ≥ 70° are described. Figure 5 presents the plastic deformation of Ti at θ = 80°. When the strain is up to 6.3% (Figure 5a), the cross-slip of the prismatic <*a*> and pyramidal <*a*> dislocation occurs at the free surface. Meanwhile, a $\left\{10\bar{1}1\right\}$ twin embryo forms through the expansion of $\left\{10\bar{1}1\right\}$ pyramidal <*a*> dislocation. In 2016, Serra et al. proposed a nucleation mechanism for $\left\{10\bar{1}1\right\}$ twin by atomistic simulations. It was indicated that the dissociation of a screw <*a*> dislocation can transform to a twin embryo [13]. Kou et al. observed the real-time $\left\{10\bar{1}1\right\}$ twinning process at the crack tip of Ti by transmission electron microscope (TEM) [14]. A twin embryo is caused by the glide on the $\left\{10\bar{1}1\right\}$ plane of emitted lattice dislocations. At the strain of 6.34%, both $\left\{10\bar{1}1\right\}$ twin, BSFs and fcc-Ti are observed in Figure 5b. In the magnified view of Figure 5d,e, it is noticed that a profuse BSF appears in the interior of $\left\{10\bar{1}1\right\}$ twin. Because of the interaction of BSF with $\left\{10\bar{1}1\right\}$ twin boundary (TB), one-layer sessile steps (b_1_) are left on the TB. Kou et al. revealed that the sessile steps can act as stable sources for the nucleation of b_2_ twinning dislocations, while the $\left\{10\bar{1}1\right\}$ twinning is accomplished by the glide of b_2_ [14]. Moreover, the accumulation of BSFs will induce the formation of fcc-Ti as shown in Figure 5d.

Figure 6a shows the plastic deformation of Mg at β = 70°. The $\left\{10\bar{1}1\right\}$ twin embryo nucleate directly at the free surface when the strain is 5.48% (Figure 6(a1)). They propagate with the increase of the strain, which is accompanied by the formation of BSFs. When the strain is up to 5.56%, the BP transformation occur inside the $\left\{10\bar{1}1\right\}$ twin region, and the $\left\{10\bar{1}1\right\}-\left\{10\bar{1}2\right\}$ double twin-like crystalline reorientation is observed (Figure 6(a2)). This double twin-like crystalline reorientation has been described by Zu et al. [56] in 2018, which is consistent with the result in this simulation. Figure 6b presents the deformation of Mg at β = 90°. In Figure 6(b1), the prismatic dislocation nucleates at the surface first at a strain of 5.93%. Subsequently, the $\left\{10\bar{1}1\right\}$ twin embryo nucleates at the intersection of prismatic dislocation and free surface. When the strain increases to 5.94%, BSF is found in the twin region as marked in Figure 6(b2).

The nucleation of the $\left\{10\bar{1}1\right\}$ twins caused by multiple slip of <*a*> dislocations is observed in the simulations of Mg and Ti samples in Figure 5 and Figure 6b. Specially, the $\left\{10\bar{1}1\right\}$ twins can nucleate directly at free surface in Mg under several loading conditions, which is related to the smaller CRSS value of the $\left\{10\bar{1}1\right\}$ twin for Mg (65MPa). It is worth noting that BSFs are popularly observed inside $\left\{10\bar{1}1\right\}$ twins in Mg and Ti. Both ends of the BSFs are connected to the twin boundaries, and BSFs grow with the migration of TBs. Several experiments and simulations have revealed the existence of high density BSFs inside $\left\{10\bar{1}1\right\}$ twins [20,57,58], including the hierarchical contraction nanotwins-stacking faults (CTWSFs) structure described by Peng et al. in 2019 [20]. Our calculation indicates that the Schmid factors of BSFs inside the $\left\{10\bar{1}1\right\}$ twin region are larger than those in the matrix, which are greater or equal to 0.402 when θ or β ≥ 70°. Thus BSFs form in $\left\{10\bar{1}1\right\}$ twins because of the high Schmid factor. The BSFs within twins are expected to play similar roles as TBs in impeding dislocation slips [58].

### 3.4. Double Twins (DTs)

Two types of DTs are observed in our simulation, $\left\{10\bar{1}1\right\}-\left\{10\bar{1}2\right\}$ DT in Mg when θ ≥ 70° and β = 70°, $\left\{11\bar{2}1\right\}-\left\{11\bar{2}2\right\}$ DT in Ti at θ = 15° and β = 10°.

$\left\{10\bar{1}1\right\}-\left\{10\bar{1}2\right\}$ DT in Mg when β = 70° is shown in Figure 6(a2), which is consistent with the $\left\{10\bar{1}1\right\}-\left\{10\bar{1}2\right\}$ double twin-like crystal reorientation described by Zu et al. [56]. After the nucleation of a primary $\left\{10\bar{1}1\right\}$ twin, a secondary $\left\{10\bar{1}2\right\}$ twin follows, combining the primary twin into a stable $\left\{10\bar{1}1\right\}-\left\{10\bar{1}2\right\}$ DT structure. A typical grain boundaries (GB) structure is identified between the parent and the secondary twin. Specially, the interface between the primary $\left\{10\bar{1}1\right\}$ twin and the secondary $\left\{10\bar{1}2\right\}$ twin shows a zigzag appearance, consisting of co-existing $\left\{10\bar{1}2\right\}$ TB and BP interface (Figure 6(a2)). This $\left\{10\bar{1}1\right\}-\left\{10\bar{1}2\right\}$ double twin-like crystal reorientation is also observed in Mg when θ ≥ 70°, which is the same as that of β = 70°.

Beside, $\left\{10\bar{1}1\right\}-\left\{10\bar{1}2\right\}$ DTs in Mg are observed under the tension stress near *a*-axis, $\left\{11\bar{2}1\right\}-\left\{11\bar{2}2\right\}$ DTs in Ti are observed at θ = 15° and β = 10° under the tension stressnear *c*-axis. The $\left\{11\bar{2}1\right\}-\left\{11\bar{2}2\right\}$ DTs have been observed by Gong et al. via high-resolution transmission electron microscope (HRTEM) [59]. Figure 7 presents the initial plastic deformation of Ti at θ = 15°. In Figure 7a, the basal <*a*> dislocation nucleates first at the free surface, then the $\left\{10\bar{1}1\right\}\langle 11\bar{2}3\rangle$ pyramidal dislocations follow. With the increase of strain, a SB grows (Figure 7b) and extends across the sample (Figure 7c) through the movement of $\left\{\bar{1}2\bar{1}1\right\}$ pyramidal partial dislocations. Afterwards, the $\left\{11\bar{2}2\right\}$ twin nucleates at the intersection of $\left\{\bar{1}2\bar{1}1\right\}$ interface and the free surface, and the $\left\{11\bar{2}1\right\}-\left\{11\bar{2}2\right\}$ double twin forms (Figure 7d). According to the geometrical analysis in Figure 7e, the Schmid factor of $\left\{11\bar{2}2\right\}$ twin within the $\left\{11\bar{2}1\right\}$ twin (f = 0.305) is larger than the value in the matrix (f = 0.282) at θ = 15°. Therefore, the secondary $\left\{11\bar{2}2\right\}$ twin forms because of the higher Schmid factor and the concentrated stress at the intersection of $\left\{11\bar{2}1\right\}$ interface and free surface. When β = 10°, $\left\{10\bar{1}1\right\}-\left\{10\bar{1}2\right\}$ DT is also observed in Ti. The preferred secondary twinning modes in hcp metals would relax the local stress and strain concentration associated with the primary region.

### 3.5. Phase Transformation

For the stress-induced phase transformation, it is hard to occur in Mg and Zr at low temperatures. In Ti single crystals, phase transformation from hcp to fcc structure is observed at θ = 0°, θ ≥ 70°, and β = 15°–58.6°. In our simulation, the hcp-fcc phase transformation is accomplished via successive glide of basal partial dislocations, which has been observed in previous studies [28,54,60].

As shown in Figure 4d, the fcc-Ti is observed when θ = 0° because of the accumulation of BSFs. Figure 8 presents the detail of the hcp-fcc transformation. The terraced interface is shown in Figure 8a, which consists of BP interfaces, $\left\{10\bar{1}2\right\}$ TB, and interfacial defects. Two types of interfacial disconnections in BP interface are found, indicated as two-layer steps and one-layer steps. In Figure 8b, the basal partial dislocations emit at the interfacial disconnections and expand to the free surface, with the BSFs left. In Figure 8c, the fcc-Ti forms because of the successive emission of basal partial dislocations and the accumulation of BSFs. It is indicated that the orientation relationship between the hcp and fcc phases is {0001}_hcp_||{111}_fcc_ and $\langle 11\bar{2}0\rangle$_hcp_ ||〈110〉_fcc_. The hcp-fcc transformation induced by the accumulation of BSFs is also observed in the interior of $\left\{10\bar{1}1\right\}$ twin as shown in Figure 5d.

When β = 35.9° (Figure 9a), the fcc-Ti induced by the accumulation of BSFs is observed related to the dissociation of $\left\{10\bar{1}1\right\}$ partial dislocations (Figure 9(a1)). With increase in the strain, the BP interface and SB appear near the fcc-Ti (Figure 9(a2)). When β = 58.6° (Figure 9b), the fcc-Ti is observed to nucleate directly at the free surface because of the higher Schmid factor of the basal slip, which is also induced by the accumulation of BSFs.

## 4. Discussion

As we know, the plastic deformation mechanisms of Mg, Zr, and Ti show a distinct difference with each other. It is closely associated with the crystal structure, a variety of the *c*/*a* ratio, and etc. The CRSS is related to the crystal structure, which describes the critical stress for plastic deformation to occur on a given deformation mode. Different metals have different CRSS values for one given slip system or twinning mode. We can analyze the ease or complexity of the activation of one deformation mode by comparing the CRSS values of different metals. The basal <*a*> slip is preferred in Mg while the prism <*a*> slip is favored in Ti and Zr, which are determined by the *c*/*a* ratio and reflected in the magnitude of the CRSS values. The basal <*a*> dislocation has a much lower CRSS value (5 MPa) than that of other slip systems (> 90 MPa) in Mg [49]. The prismatic <*a*> dislocation is the most active defect with the minimum value for Zr (15 MPa) and Ti (60 MPa) [53,54]. The CRSS of the $\left\{10\bar{1}2\right\}$ twin in Mg (30 MPa) is also much lower than that of Zr (165 MPa) and Ti (260 MPa) [49,53,54]. The lower CRSSs of slip and twinning in Mg result in the strength of Mg that is much lower than that of Ti and Zr. Moreover, more types of plastic deformation mechanisms, including cross-slip, twining, and phase transformation, can be simultaneously activated in Ti, inducing the higher strength and ductility of Ti and its alloys.

Because of the asymmetric and anisotropy crystallographic structure, the orientation effect is very important for the deformation of hcp metals. When the load is applied in different direction, the yield stress is rather changeable as the low symmetry and the limited slip systems of hcp metals. The Schmid factors are used to analyze the orientation effect, which are determined by the loading directions and *c*/*a* ratio. For a given deformation mechanism, the Schmid factors for different hcp metals (Mg, Zr or Ti) have a tiny difference. The slip system with a larger Schmid factor has priority to be activated during the deformation. Generally, comprehensive analysis of Schmid factors and CRSSs of slips is almost matched with the consequence of the deformation mechanism in our simulation. For an example, the basal <*a*> slip occurs prior to the pyramidal <*c* + *a*> slip when β = 10° in Mg and Ti, although the Schmid factor of the $\left\{10\bar{1}1\right\}\langle 11\bar{2}3\rangle$ pyramidal slip (f~0.470) is much higher than that of the basal <*a*> slip (f = 0.125). This is due to the higher CRSS for the <*c* + *a*> slip than the basal <*a*> slip at low temperatures. The combined effect of CRSS and the Schmid factor determines the prior activated slip system. Moreover, when the loading is applied near basal plane, the $\left\{10\bar{1}1\right\}$ twins are formed in Mg and Ti because of its higher Schmid factors. But no $\left\{10\bar{1}1\right\}$ twin appear in Zr in our simulation. This is reasonable because Zr has a high CRSS value of $\left\{10\bar{1}1\right\}$ twin at low temperatures. Many researchers reported that the $\left\{10\bar{1}1\right\}$ twin can only form at elevated temperatures in Zr [17,18].

Based on the systematic analysis of the deformation mechanisms in hcp metals, the affecting factors for the plasticity of hcp materials are discussed. In order to improve the mechanical properties of hcp materials, several methods can be applied according to the above analysis. At first, alloying with some elements might lead to changes in the CRSSs of the different deformation modes with respect to pure hcp metals. Many researchers have observed the alloying effect on the CRSS in hcp metals. Qian et al. have reported that the addition of Li to Mg increases the CRSS for basal slip, while the CRSS for prism slip is lowered [61]. Ojha et al. discussed the effect of Nb and Ta contents on the CRSS for the slip, twinning, and phase transformation in Ti-based alloys [62]. Westlake et al. proposed that the CRSS of Zr was increased as much as 500% by the addition of H in the form of finely dispersed hydride needles [63]. Besides the CRSSs, the crystallographic texture related with the orientation effect can effectively affect the deformation mechanisms. For example, the formation of basal texture as well as the decrease of the grain size can increase the formability of Mg [64]. Sattari et al. put forward that the crystallographic texture of the pressure tube has a key role in its in-reactor behavior such as creep and irradiation growth [65]. A previous study has shown that micro-texture has a strong influence on both advancing and inhibiting the growth of adiabatic shear band in hcp metals, and the control of micro-texture might offer the possibility of new alloys with higher impact strength [66].

## 5. Conclusions

The plastic deformation mechanisms of Mg, Zr, and Ti single crystals with different orientations are systematically investigated by using molecular dynamics simulations. For different orientation samples with θ or β ranging from 0° to 90°, different plastic deformation mechanisms are exhibited. By comparing the deformation mechanisms of different hcp metals and analyzing the orientation effect on deformation behaviors, the affecting factors for the plasticity of hcp metals are analyzed.
(1)The slips dominate the plasticity of hcp single crystals. The basal slip is preferred in Mg while the prism slip is favored in Ti and Zr, which are characterized by c/a ratio and the CRSS values. The slip mechanisms observed in our simulation match well with the comprehensive analysis of the Schmid factor and CRSS of slip systems.(2)When the basal and prismatic slips are restricted, the twinning is activated. The $\left\{10\bar{1}2\right\}$ twinning is popularly observed in Mg, Zr, and Ti single crystals because of its low CRSS, which is usually accompanied with the BP transformation. The $\left\{10\bar{1}1\right\}$ twins form in Mg and Ti, but not in Zr. The $\left\{1\bar{2}11\right\}$ type II twins (SBs) appear in Mg, Zr, and Ti. Moreover, the $\left\{10\bar{1}1\right\}-\left\{10\bar{1}2\right\}$ DT in Mg and $\left\{11\bar{2}1\right\}-\left\{11\bar{2}2\right\}$ DT in Ti are observed, responsible for relaxing the local stress and strain concentration associated with the primary twin.(3)The stress-induced phase transformation from hcp to fcc structure in Ti is observed, which is achieved by the accumulation of BSFs. For samples with different orientations, the fcc phase can be nucleated at the defects or the free surface via the activation and successive glide of basal partial dislocations.(4)The lower CRSSs of slip and twinning in Mg result in the strength of Mg is much lower than that of Ti and Zr. Moreover, there are more types of plastic deformation mechanisms in Ti than in Mg and Zr. Multiple deformation mechanisms coordinate with each other, resulting in high strength and ductility of Ti.

Based on the comprehensive analysis of the deformation mechanisms in hcp single crystals, the strengthening mechanisms for the hcp materials are discussed. Alloying with the elements might lead to the change of the CRSS with respect to pure hcp metals, resulting in the improvement of the mechanical properties. Moreover, the processing technique related with the orientation effect can effectively affect the deformation mechanisms, which can be implemented to design novel hcp materials.

## Figures and Tables

**Figure 1 materials-14-00733-f001:**
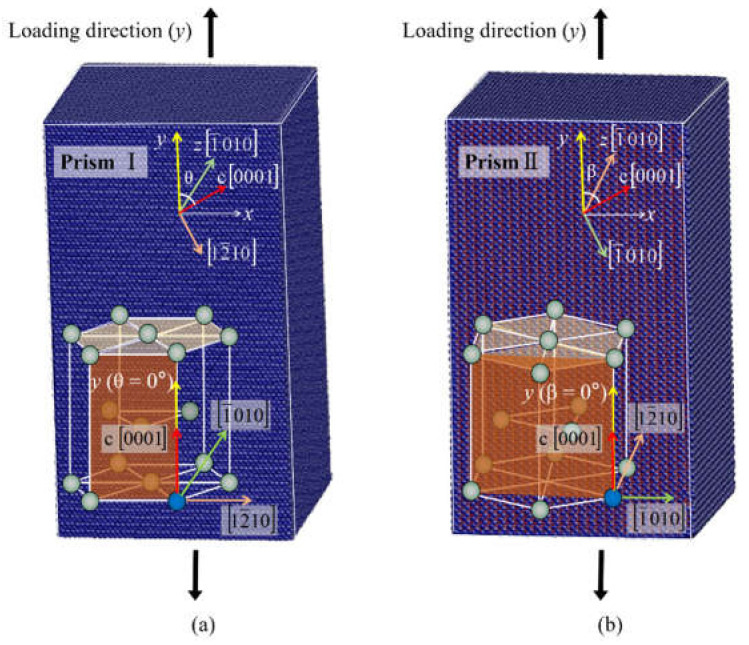
Simulation models. (**a**) model A and (**b**) model B. The *y*-axis indicates the loading direction.

**Figure 2 materials-14-00733-f002:**
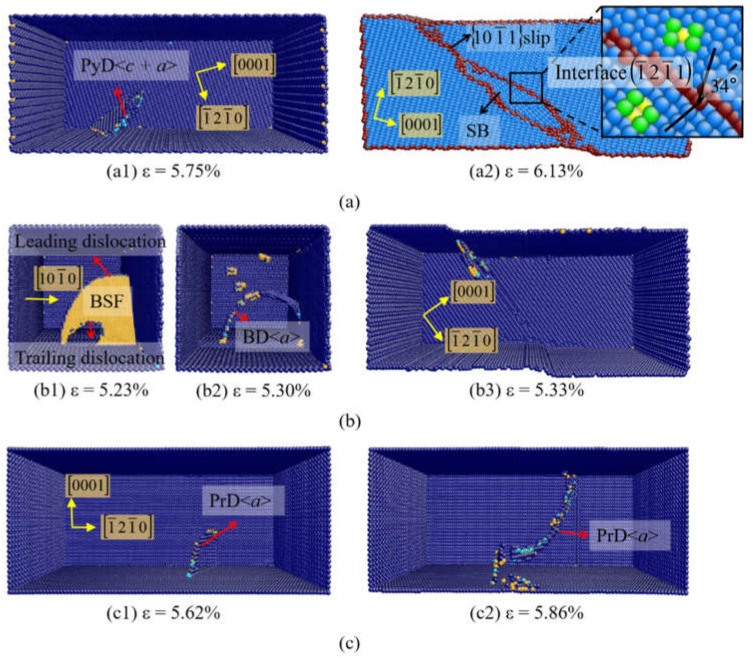
Atomic configurations of Zr at (**a**) θ = 15°, (**b**) θ = 32.1°, (**c**) θ = 90°.

**Figure 3 materials-14-00733-f003:**
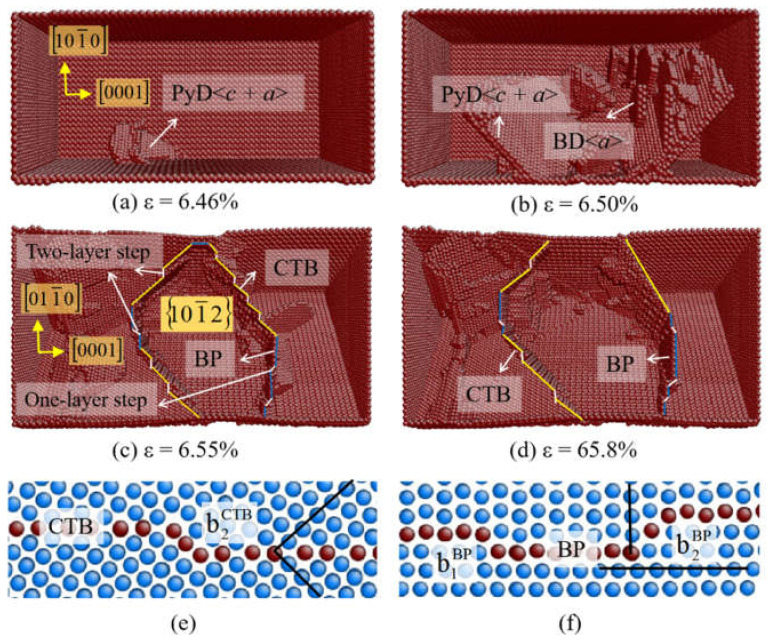
(**a**–**d**) Atomic configurations of Zr at θ = 0°. (**e**,**f**) The magnified atom configurations of the terraced interface consists of $\left\{10\bar{1}2\right\}$ CTB and BP interfaces, together with the interface defects.

**Figure 4 materials-14-00733-f004:**
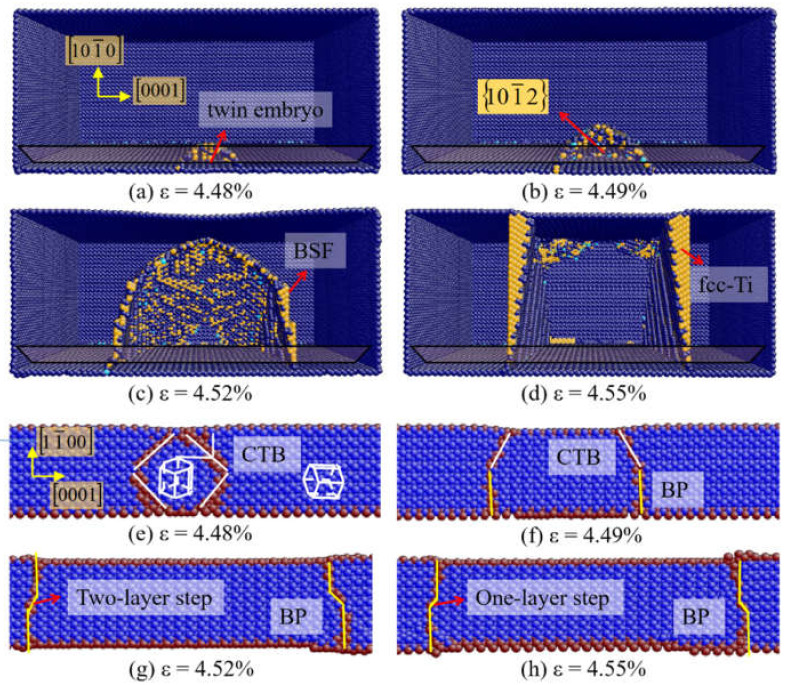
(**a**–**d**) Atomic configurations of Ti at θ = 0°. (**e**–**h**) The nucleation and growth process of $\left\{10\bar{1}2\right\}$ twin viewed along the $\left[1\bar{2}10\right]$ direction.

**Figure 5 materials-14-00733-f005:**
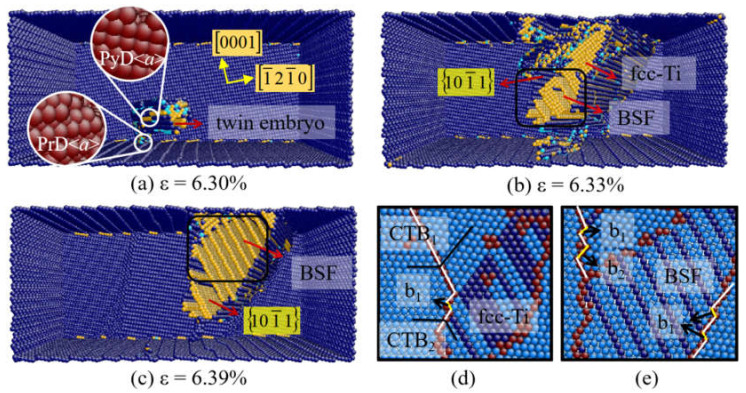
(**a**–**c**) Atomic configurations of Ti at θ = 80°. (**d**,**e**) Magnified views of defects.

**Figure 6 materials-14-00733-f006:**
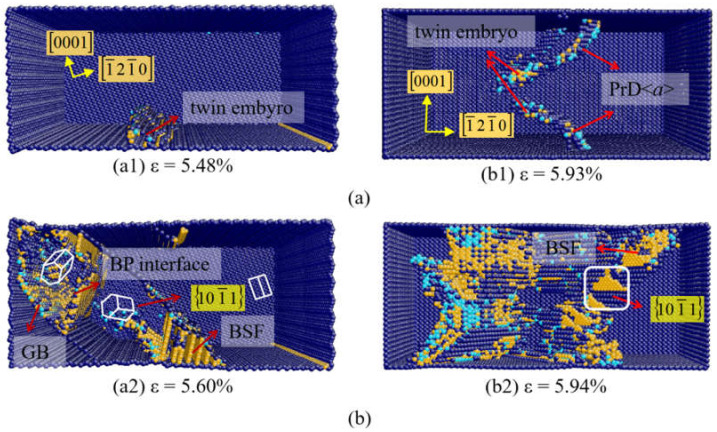
Atomic configurations of Mg at (**a**) β = 70°, (**b**) β = 90°.

**Figure 7 materials-14-00733-f007:**
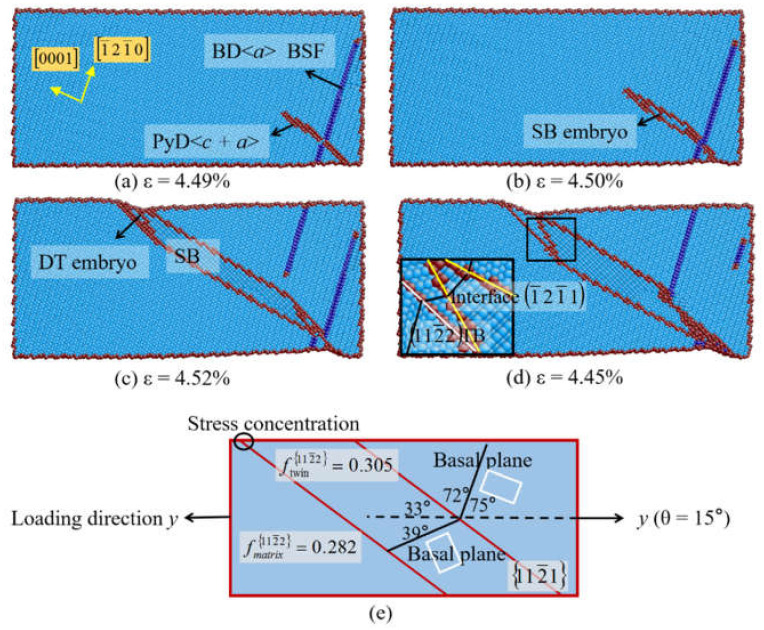
(**a**–**d**) Atomic configurations of Ti at θ = 15°. (**e**) The geometrical analysis of $\left\{11\bar{2}1\right\}-\left\{11\bar{2}2\right\}$ DT.

**Figure 8 materials-14-00733-f008:**
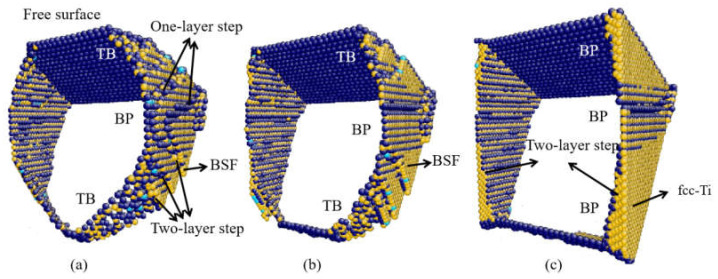
(**a**–**c**) fcc-Ti transformation at θ = 0°.

**Figure 9 materials-14-00733-f009:**
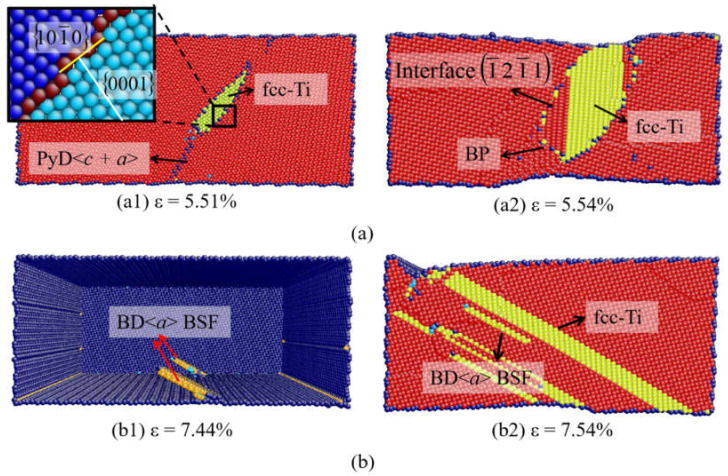
fcc-Ti transformation at (**a**) β = 35.9°and (**b**) β = 58.6°.

**Table 1 materials-14-00733-t001:** Plastic deformation mechanisms of hcp single crystals under tension with different orientations (θ).

θ (°)	Mg [26] (c/a = 1.623)	Zr (*c*/*a* = 1.593)	Ti (*c*/*a* = 1.588)
0 (*c*-axis)	PyD <*c* + *a*>, BD <*a*>, BSF, T3, BP	PyD <*c* + *a*>, BD <*a*>, BSF, T3, BP	T3, BP, BSF, fcc-Ti
10	SB	PyD <*c* + *a*>, SB	PyD <*c* + *a*>, SB
15	SB	PyD <*c* + *a*>, SB	BD <*a*>, BSF, PyD <*c* + *a*>, SB, DT
32.1	SB	BD <*a*>, BSF	BD <*a*>, BSF
43.3	PrD <*a*>, BD <*a*>, BSF	PrD <*a*>, BD <*a*>, BSF	BD <*a*>, BSF
51.6	PrD <*a*>, BD <*a*>, BSF	PrD <*a*>, BD <*a*>, BSF	BD <*a*>, BSF
70	PrD <*a*>, T1, BSF, BP, GB	PrD <*a*>	PrD <*a*>, PyD <*a*>, T1, BSF, fcc-Ti
75	PrD <*a*>, T1, BSF, BP, GB	PrD <*a*>	PrD <*a*>, PyD <*a*>, T1, BSF, fcc-Ti
80	T1, BSF, BP, GB	PrD <*a*>	PrD <*a*>, PyD <*a*>, T1, BSF, fcc-Ti
90	T1, BSF, BP, GB	PrD <*a*>	PrD <*a*>, PyD <*a*>, T1, BSF, fcc-Ti

Note: θ is the angle between the loading direction (*y*) and the *c*-axis in model A. Results in Mg, Zr and Ti samples are shown. The dislocation is in orange (including basal <*a*> dislocation (BD <*a*>), basal stacking fault (BSF), prismatic <*a*> dislocation (PrD <*a*>), pyramidal <*c* + *a*> dislocation (PyD <*c* + *a*>), shear band (SB), and the cross-slip of PrD <a> and PyD <a>). The lattice reorientation is in blue (including $\left\{10\bar{1}1\right\}$ twin (T1), $\left\{10\bar{1}2\right\}$ twin (T3), $\left\{11\bar{2}1\right\}-\left\{11\bar{2}2\right\}$ double twin (DT), basal/prismatic transformation (BP)). The phase transformation is in green (fcc-Ti).

**Table 2 materials-14-00733-t002:** Plastic deformation mechanisms of hcp single crystals under tension with different orientations (β).

β (°)	Mg (*c*/*a* = 1.623)	Zr (*c*/*a* = 1.593)	Ti (*c*/*a* = 1.588)
0 (*c*-axis)	PyD <*c* + *a*>, BD <*a*>, BSF, T3, BP	PyD <*c* + *a*>, BD <*a*>, BSF, T3, BP	T3, BP, BSF, fcc-Ti
10	BD <*a*>, BSF, PyD <*c* + *a*>, SB	PyD <*c* + *a*>, SB	BD <*a*>, BSF, PyD <*c* + *a*>, SB, DT
15	BD <*a*>, BSF, PyD <*c* + *a*>, SB	PyD <*c* + *a*>, BD <*a*>, BSF	BD <*a*>, BSF, fcc-Ti
28.6	BD <*a*>, BSF	PyD <*c* + *a*>, BD <*a*>, BSF	BD <*a*>, BSF, fcc-Ti
35.9	BD <*a*>, BSF	PyD <*c* + *a*>, BD <*a*>, BSF	PyD <*c* + *a*>,fcc-Ti, BP
47.4	BD <*a*>, BSF	BD <*a*>, BSF, PrD <*a*>, PyD <*c* + *a*>	BD <*a*>, BSF, fcc-Ti
55.4	BD <*a*>, BSF	BD <*a*>, BSF, PrD <*a*>, PyD <*c* + *a*>	BD <*a*>, BSF, fcc-Ti
58.6	BD <*a*>, BSF	BD <*a*>, BSF, PrD <*a*>, PyD <*c* + *a*>	BD <*a*>, BSF, fcc-Ti
70	T1, BSF, BP, GB	PyD <*c* + *a*>, PrD <*a*>	PrD <*a*>, BD <*a*>, BSF
75	PrD <*a*>, T1, BSF	PyD <*c* + *a*>, PrD <*a*>	PrD <*a*>, BD <*a*>, BSF
80	PrD <*a*>, T1, BSF	PyD <*c* + *a*>, PrD <*a*>	PrD <*a*>
90	PrD <*a*>, T1, BSF	PyD <*c* + *a*>, PrD <*a*>	PrD <*a*>

Note: the description of the color is consistent with the note of Table 1.

**Table 3 materials-14-00733-t003:** Schmid factors (f) of slip systems for hcp single crystal with different θ.

Loading Direction (θ)	BD <*a*>	PrD <*a*>	PyD <*c* + *a*> $\left\{10\bar{1}1\right\}\langle 11\bar{23}\rangle$			PyD <*c* + *a*> (SB) $\left\{1\bar{2}11\right\}\langle 11\bar{2}6\rangle$		
			Mg	Zr	Ti	Mg	Zr	Ti
0	0	0	**0.401**	**0.404**	**0.404**	0.281	0.286	0.286
15	**0.25**	0.029	0.493	**0.493**	0.493	**0.451**	0.453	0.454
32.1	**0.450**	0.122	0.468	0.466	0.466	**0.495**	0.494	0.493
51.6	**0.487**	**0.266**	0.288	0.284	0.283	0.469	0.464	0.464
70	0.321	**0.382**	0.436	0.435	0.434	0.481	0.480	0.484
90	0	**0.433**	0.400	0.405	0.405	0.282	0.286	0.287

Note: The initially nucleated dislocations of each angles are marked in red.

**Table 4 materials-14-00733-t004:** Schmid factors (f) of slip systems for hcp single crystal with different β.

Loading Direction (β)	BD <*a*>	PrD <*a*>	PyD <*c* + *a*> $\left\{10\bar{1}1\right\}\langle 11\bar{23}\rangle$			PyD <*c* + *a*> (SB) $\left\{1\bar{2}11\right\}\langle 11\bar{2}6\rangle$		
			Mg	Zr	Ti	Mg	Zr	Ti
0	0	0	**0.401**	**0.404**	**0.404**	0.281	0.286	0.286
10	**0.125**	0.013	0.469	**0.470**	0.470	0.388	0.388	0.392
28.6	**0.365**	0.099	0.442	**0.440**	0.438	0.470	0.470	0.472
55.4	**0.399**	0.293	0.394	0.389	0.390	0.388	0.386	0.387
70	0.278	**0.382**	**0.480**	**0.479**	0.477	0.383	0.383	0.386
90	0	**0.433**	0.401	**0.405**	0.405	0.211	0.214	0.216

Note: The initially nucleated dislocations of each angles are marked in red.

## Data Availability

Data is contained within the article.

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
