# Peer review of "The Plastic Deformation Mechanisms of hcp Single Crystals with Different Orientations: Molecular Dynamics Simulations"

_materials, 2021, doi:10.3390/ma14040733_

Round 1

Reviewer 1 Report

The plastic deformation mechanisms of hcp single crystals with different orientations: Molecular dynamics simulations
Z. Ma, X. Tang, Y. Mao, and Y. Guo

A thorough investigation of several hcp crystals undergoing different
orientations is described in this paper. The authors show careful
molecular dynamics (MD) simulations of hcp crystals and described them
well. Their conclusions are in agreement with the results they
observe. In particular, they observe that the main slip system that is
activated during MD simulations is the basal one in magnesium, and the
prismatic one for titanium and zirconium.

They also study and observe twinning using MD. Careful figures and
analysis are shown to describe appearing twins during
simulations. Lastly, they observe phase transformations during
simulations. Figures describe the mechanism with a lot of care.

Overall, this paper is interesting. I would recommend it for
publication. However, several sentences need to be corrected and the
English in the text needs to be improved.

A few sentences below need to be rewritten because they are not clear.

``The twinning can be activated when the basal and prismatic slips are restricted.''

``the activation of pyramidal <c+a> slips are possible'' -> are to is

``As we know, the plastic deformation mechanisms of hcp metals, including Mg, Zr
and Ti, show a distinct different with each other.''

``Generally, comprehensive analysis of Schmid factors and CRSSs of slips is 404
almost match the consequence of the deformation mechanism in our simulation.''

A few sentences have extra ``the'' like for instance:
``In general, the slip is always along the lattice close-packed direction on the close-packed plane.'' The slip -> Slip

Reviewer 2 Report

In the manuscript entitled: "The plastic deformation mechanisms of hcp single crystals with different orientations: Molecular dynamics simulations" the Authors investigated the deformation mechanisms of different hcp single crystals, such as Mg, Zr and Ti, with different c/a ratios under different loading conditions. The Authors compared the deformation mechanisms of different hcp metals and analyzed the orientation effect on deformation behaviors. The affecting factors for the plasticity of hcp metals are also discussed.

Manuscript is written well and the results are clearly presented. In my opinion paper is interesting and worth publication in Materials after minor revision.

Specific comments:

1. Please delete "1;" from the keywords.
2. Each highlighting colour should be described in the footnotes of both Table 1 and Table 2 (e.g. (PyD <c+a>) - highlighted in ... etc.) rather than in the text.
3. Why some of the values presented in the Tables 3 and 4 are highlighted in yellow or marked in red? It should be explained in the above mentioned tables footnotes.
4. Line 318, Fig. 5 Caption - "(e, f)" should be replaced by "(d, e)"
5. Some newest References should be cited in the Introduction (e.g. Metals, 2020, 10, 1134; doi:10.3390/met10091134; Materials Research, 2020, 23(1), DOI: 10.1590/1980-5373-mr-2019-0638; Chinese Physics B, 2020, 29(4),046201).
